# All Are in a Drought, but Some Stand Out: Multivariate Analysis in the Selection of Agronomic Efficient Popcorn Genotypes

**DOI:** 10.3390/plants11172275

**Published:** 2022-08-31

**Authors:** Jhean Torres Leite, Antônio Teixeira do Amaral Junior, Samuel Henrique Kamphorst, Valter Jário de Lima, Divino Rosa dos Santos Junior, Uéliton Oliveira Alves, Valdinei Cruz Azeredo, Jacymara Lopes Pereira, Rosimeire Barboza Bispo, Katia Fabiane Medeiros Schmidt, Flávia Nicácio Viana, Alexandre Pio Viana, Henrique Duarte Vieira, Helaine Christine Cancela Ramos, Rodrigo Moreira Ribeiro, Eliemar Campostrini

**Affiliations:** Laboratory of Plant Breeding, Center of Agricultural Science and Technology, Darcy Ribeiro State University of Northern Rio de Janeiro, Av. Alberto Lamego, 2000, Campos dos Goytacazes 28013-602, RJ, Brazil

**Keywords:** genetic resources, drought, climate changes

## Abstract

The search for productive germplasm adapted to adverse conditions is an important action to mitigate the harmful effects of climate change. The aim was to identify the yield potential of 50 popcorn inbred lines grown in field conditions, in two crop seasons (CS), and under contrasting water conditions (WC). Morphoagronomic, physiological, and root system traits were evaluated. Joint and individual analyses of variance were performed, in addition to the multivariate GT bip-lot analysis. Expressive reductions between WC were observed in 100-grain weight (100 GW), popping expansion (PE), grain yield (GY), expanded popcorn volume per ha (EPV), row number per ear (RNE), plant height (PH), relative chlorophyll content (SPAD), and nitrogen balance index (NBI). It was found that the SPAD, 100 GW, GY, PE, and grain number per ear (GNE) traits had the most significant impact on the selection of genotypes. Regardless of WC and CS, the ideal lines were L294 and L688 for PE; L691 and L480 for GY; and L291 and L292 for both traits. SPAD, 100 GW, and GNE can contribute to the indirect selection. Our work contributes to understanding the damage caused by drought and the integration of traits for the indirect selection of drought-tolerant popcorn genotypes.

## 1. Introduction

Resilience to environments with limited natural resources is an important feature to mitigate the harmful effects of climate change [1,2]. Due to the increase in CO_2_ concentration and temperature, the scarcity and irregularity of rainfall represents the climactic event that most affects agriculture [3]. Drought causes numerous losses in world agriculture and can cause damage to cereal production in proportions greater than 50% [4]. In popcorn, extreme water shortages can cause damage that compromises up to 60% of grain yield [5]. In this sense, identifying genotypes with high yield potential and high adaptive capacity to adverse environments is a challenge for breeders, physiologists, phytotechnologists, and biotechnologists who work with abiotic stresses [6,7].

The main way to identify the secondary traits associated with adaptation to abiotic stresses and high yield potential is to evaluate the germplasm of the target species [7,8]. The screening of genetic resources allows the development of cultivars with different expressions and superior performance in response to water restriction [9]. In this sense, for popcorn, the integration of secondary characteristics related to grain yield and the potential of expansion creates new horizons of possibilities for the development of cultivars [10,11,12]. Under water deficit conditions, plants are subjected to numerous morphophysiological, biochemical, and molecular changes at different intensity levels [13,14]. At the morphoagronomic level, for the evaluation of germplasm aiming to obtain higher grain yield (GY) under water deficit in the soil, a shorter interval between male and female flowering [15], a deep root system [16], late leaf senescence (higher stay green index) [17], effective control of stomatal conductance/transpiration in association with lower leaf canopy temperature [18], an increase in prolificacy and a smaller number of branches of the tassel [11], as well as larger stem diameter and a higher net photosynthetic rate [12], are considered as the main secondary traits associated with GY.

Popcorn is a crop of great economic importance considering that this species has high added value, being marketed at a higher price than standard corn [19]. Popcorn is a food that comes from the expansion of the grain and is very nutritious, having high levels of antioxidant molecules and good content of proteins, fibers, vitamins, sugars, and fats [20]. Popcorn consumption is increasing in Brazil, being sold in several establishments destined exclusively for human consumption. National production is not sufficient to supply the domestic market, requiring imports, especially from the United States of America and Argentina [21,22], in addition to the fact that the grain is cultivated in the second crop season of the year, period of higher incidence of water instability [23]. This scenario, associated with frequent global climate changes, drives the development of more yielding popcorn cultivars adapted to drought.

The State University of Northern Rio de Janeiro (Universidade Estadual do Norte Fluminense “Darcy Ribeiro”—UENF) has been developing a popcorn breeding program for drought adaptation that is unique in Brazil [9,10,11,12,24,25,26]. In the pioneering studies, germplasm derived from Brazilian material was used; however, to enhance the prospect of promising results, a new panel of lines was developed by the institution, originating from popcorn populations in Latin America, with adaptation to temperate and tropical climates. In this context, through multivariate analyses, the aim was to select promising popcorn lines and identify yield components that could help in the indirect selection of genotypes under water-limited soil conditions.

## 2. Results

### 2.1. Analysis of Variance, Estimates of Means, and Impact of Water Limitation on the Morphoagronomic, Physiological, and Morphological Traits of the Root System

In the joint analysis between crop seasons (CS), only the number of brace roots (NBR) and density of brace roots (DBR) did not express a significant difference at 5% of probability (Table 1). For the water conditions (WC), except for the angle of brace root (ABR) and density of crown roots (DCR), all other traits showed high significance (*p* ≤ 0.001). For the Genotype (G), all traits showed a significant difference (*p* ≤ 0.001). All double interactions, namely, CS*WC, CS*G, and WC*G, showed significance (*p* ≤ 0.001), except for the NBR and DBR traits concerning the CS*WC interaction (Table 1). All evaluated traits were influenced by the CS*WC*G interaction (Table 1).

In the 2020 crop season (24.5% reduction in irrigation water availability), expressive decreases (>15%) were observed among water conditions (WC), given the comparison between the averages found in WS concerning WW, based on the values of the 100 GW (15.50%), PE (21.45%), GY (59.98%), EPV (65.89%), RNE (18.30%), PH (17.55%), and SPAD (49.11%) (Figure 1). In the 2021 crop season, PE (15.35%), GY (36.81%), EPV (45.70%), SPAD (56.34%), and NBI (26.62%) had the expressive reductions (>15%) (Figure 1). The ANT and FLV showed an increase in the WS condition concerning WW, in the magnitudes of 16.60% and 0.50%, in the 2020 crop season, and of 37.57 and 15.79%, in the 2021 crop season, respectively (Figure 1).

Inexpressive reductions (<15%) were observed in 2020 CS for the agronomic traits, GNR (11.22%), ED (9.40%), EL (11.32%), and EH (10.81%); and in RNE (1.10%), GNR (6.51%), ED (6.40%), EL (4.36%), EH (3.77%), PH (3.17%), and SD (10.93%), in 2021 CS (reduction of 18% in soil water availability through irrigation) (Figure 1). On the other hand, there was a slight increase in SD (5.14%) and NBI (0.08%) in 2020 CS, and in Fv/Fm (1.48%) in 2021 CS (Figure 1).

The root traits had inexpressive reductions (<15%) for NBR (8.37%), NCR (3.45%), and DBR (1.09%) in the 2020 CS; and for ABR (8.31%), ACR (5.91%), NBR (10.78%), NCR (8.24%), DBR (8.61%), and DCR (5.42%) in the 2021 CS (Figure 1). The increase in the value of the ABR (6.78%), ACR (1.33%), and DCR (10.00%) stands out, given the comparison of WS concerning WW in the 2020 CS (Figure 1).

### 2.2. Multivariate Analysis—GT Biplot

The two principal components explained more than 60% of the variation, regardless of CS and WC (Figure 2). In WS, in the 2020 and 2021 crop seasons, the values were 63.73% and 60.98%, respectively; in WW, in the 2020 and 2021 crop seasons, they were 66.70% and 68.96%, respectively (Figure 2).

Regardless of WC, four distinct groups were formed in the 2020 CS in the “which-won-what” graphic space (Figure 2A,B). In the WS condition, the first group included only 100 GW, formed between the perpendicular red lines, in which the L652 (44) line stood out, located at the vertex of the polygon between the perpendicular red lines (Figure 2A). The second group was formed by GY, PE, RNE, and ED, in which the L381 (24) line was the highlight; the third group was formed only by GNE, standing out the L294 (13) line. Finally, the fourth group was formed by EL and GNR, in which the L358 (22) and L502 (36) lines stood out (Figure 2A). In the WW condition, the first group was formed by PE, emphasizing the L222 (8) line; the second consisted of 100 GW, RNE, and ED, focusing on the L381 (24) line. The third was formed by GNE and GY, emphasizing the L292 (12) and L503 (37) lines, which presented high values. The fourth and last group gathered the GNR and EL, with emphasis on the L430 (33) and L502 (36) lines (Figure 2B).

In 2021 CS, two groups were formed for the WS condition and three for the WW condition (Figure 2C,D). In the WS condition, the first group included GY, EL, GNR, and GNE, and was allowed to highlight the L291 (11) and L217 (4) lines; the second group was formed by RNE, ED, PE, and 100 GW, with emphasis on the L625 (43) line (Figure 2C). In the WW condition, the first group was formed by PE, with emphasis on the L684 (46) and L220 (6) lines; the second group gathered GY, 100 GW, ED, and RNE, with a focus on the L480 line (33); finally, the third group consisted of GNE, EL, and GNR, emphasizing the L510 (40) line (Figure 2D).

In the analysis of the “Mean vs. stability” biplot, in the WS condition, 24 and 23 lines had means higher than the general mean; 26 and 27 lines expressed values below the general average, respectively, in the 2020 and 2021 crop seasons (Figure 3A,C).

In 2020 CS, only the L294 (13) line was considered ideal, having high performance concerning the general average of the traits and greater stability, based on the extended projection of the dashed line and location of the genotype close to the arrow with the concentric circle in the graph (Figure 3A). However, the L381 (24), L292 (12), and L291 (11) lines, although they did not stand out as the closest to the ideal, showed considerable stability in the WS condition, given the graphic proximity of the genotype to the concentric circle and the extension of the dashed line (Figure 3A). In the 2021 CS, no line was considered ideal; however, L381 (24) and L625 (43) lines stood out as the closest to the ideotype, with considerable stability under WS conditions (Figure 3C).

In the WW condition, in the 2020 CS, the L292 (12) and L503 (37) lines were considered ideal, presenting high performance concerning the general average of the traits and considerable stability (Figure 3B). The L381 (24), L391 (29), and L480 (33) lines, despite not showing ideal averages, stood out with good stability and high averages for the evaluated traits (Figure 3B). In 2021 CS, no line was considered ideal; however, L480 (33), L203 (1), L691 (49), and L292 (12) lines presented good stability and means with representative magnitude values for the evaluated traits (Figure 3D).

In the analysis “Discrimitiveness vs. representativeness”, the PE characteristic showed a low discriminatory capacity, in the 2020 CS, and WS conditions (Figure 4A). However, it was highly representative, given its smaller angle concerning the axis with the arrow and concentric circle (Figure 4A).

The GY stands out, which expressed high discriminatory capacity and good representation, resulting from the extensive projection of the dashed line and the reduced angle concerning the axis containing the concentric circle (Figure 4A). The 100 GW, EL, GNR, and ED traits had high discriminatory capacity but low representative capacity. The RNE and GNE traits showed high discriminatory capacity and median representative capacity (Figure 4A). In the 2021 CS, PE and 100 GW showed the highest discriminatory capabilities; however, they had low representativeness (Figure 4C). ED stands out, which presented the highest representative capacity with high discriminatory capacity. RNE showed median discriminatory and representative capacities. The other traits—GY, EL, GNR, and GNE—showed high discriminatory and medium representative capacities (Figure 4C).

In the WW condition and the 2020 CS, the GY and GNE traits presented the highest representative and discriminatory capacities. The ED, RNE, PE, EL, and GNR traits showed high representative capacity, but low discriminatory capacity. The 100 GW presented median discriminatory and representative capacities (Figure 4B). In the 2021 CS, GY, GNE, and THE presented the highest discriminatory capacities combined with high representative capacities (Figure 4D). The other traits—ED, 100 GW, PE, EL, and GNR—showed high discriminatory capacities, however, low representative capacities (Figure 4D).

For the 2020 CS, the “Ranking Genotypes” graph in the WS condition discriminated L294 (13), L292 (12), L291 (11), L381 (24), L328 (19), and L688 (47) (Figure 5A) as the ideal lines, that is, those with greater performance and stability, as they are located closer to the concentric circle in the graph (Figure 5). In the WW condition, the highlights were for L292 (12), L391 (29), and L503 (37), respectively (Figure 5B). In 2021 CS, in the WS condition, the lines considered ideal were L273 (10), L386 (28), L381 (24), L693 (50), and L217 (4), in that order (Figure 5C); and in the WW condition, they were, hierarchically, L691 (49), L203 (1), L292 (12), L291 (11), L381 (24), and L688 (47) (Figure 5D).

## 3. Discussion

### 3.1. Genetic Variability in Different WC and CS

Genetic variability was observed between the lines for most of the evaluated traits, also noting the efficiency of the water suspension in the period of male pre-anthesis to differentiate WC. Withholding irrigation from 10 to 15 days before male flowering is widely adopted by corn breeders worldwide [10,12,17,27,28,29,30]. The use of this procedure makes it possible to reliably affirm the adaptive response of plants to the drought condition, which can be measured, above all, through the productive potential [28,29].

Due to the significant effect of CS and WC and the significant interactions with genotype (G), a differential response of the evaluated popcorn lines is expected. In the plant selection process, interactions of this nature interfere in the recommendation of cultivars for specific environments, as well as for selection gains (Hallauer et al., 2010). Regarding the presence of significance for the interactions G*CS, G*WC, and G*CS*WC, it can be inferred that, for these traits, selection under irrigated conditions or with water deficit may be effective in obtaining simultaneous genetic gains in both CS and WC. For this reason, the experiments were analyzed individually; thus, the genotype effect (G) was effectively measured in both crop seasons and water conditions.

### 3.2. Impact of Water Limitation

Regardless of the CS, the EPV was the most affected by water limitation. EPV is a trait that includes GY and PE; therefore, the sum of these effects drastically reduced the expression of EPV. The reductions in the percentages of GY of 59.98% in the 2020 crop season and 36.81% in the 2021 crop season were mainly due to the impact of water stress on the components related to the grain number per ear (GNE) in the first CS, and to the grain weight (GW100) in the second CS, which are characteristics related to the main component of grain yield, in this case, GY. In the 2020 CS, the permanent wilting point occurred earlier, i.e., close to the phenological stage, impacting the number of grains produced. In the 2021 CS, the wilting point event occurred later, in the grain filling phase, which favors understanding the most significant impact of this crop season on the grain weight, as these had already been formed. Pollen viability and zygote formation are physiological processes sensitive to soil water limitation [31], which reduces the number of grains produced. This effect may have occurred in the 2020 CS, in which the sequelae of water deficit occurred earlier and more intensely than in the 2021 CS, that is, in a stage before anthesis and, therefore, having a greater impact on GY in the 2020 CS. In turn, in the 2021 CS, whose WS effect occurred later and less intensely, the main adverse effect of water deficit was perceived in 100 GW, which impacted to a lesser degree GY since the other yield components, which are formed earlier, were less affected.

The popping expansion (PE), the main quality trait of the popcorn trade [32], was significantly affected by the imposition of the water deficit (decrease of 21.45% and 15.35% in the 2020 and 2021 crop seasons). The expansion process is associated with the presence of moisture contained in the starch granules of grain, which, when heated (≈180 °C), exerted pressure on the pericarp, whose rupture exposes the endosperm [33,34,35]. In this sense, water scarcity during grain formation can affect the physicochemical properties, interfering with the expansion capacity of the grains. Even so, morphological or chemical traits that could explain this phenomenon have not yet been recorded. In studies with popcorn carried out by [9,24,36], it is possible to observe lower reductions of PE in the magnitudes of 8.76%; 9.08% and 3.50%, respectively, between WW and WS conditions, although in the penultimate study lines and hybrids were evaluated, and in the last one, open-pollinated varieties. However, it is necessary to analyze this with caution since [10] identified an average loss of 29.19% for PE between WW and WS conditions in popcorn lines, although, in the present work, in which lines were also evaluated, this reduction was less prominent, with a magnitude of 21.45% in 2020 and 15.35% in 2021.

The period of water deficit imposition, that is, in pre-anthesis, may be related to the low reduction in PH (3.17%), EH (3.77%), and SD (10.93%). Plants close to the pre-flowering stage are at the end of vegetative growth, and at this stage, growth-related traits are less affected by limiting water conditions [37,38]. In this sense, the growth would have more significant decreases if applied to plants in the V6 to V8 stages, as presented by [39,40], in which the reductions for PH were more expressive, with magnitudes of 20% and 30%, respectively.

The expressive reductions observed in the SPAD index in both crop seasons may be due to stomatal closure, in association with high solar radiation, which, consequently, induces the formation of reactive oxygen species (ROS), which is associated with the degradation of chlorophyll molecules [41,42]. This process of formation of ROS’s can promote the inhibition of the concentration and activity of the enzymes RUBISCO and PEPcase [43]. In this sense, in both crop seasons, with the application of water deficit, significant increases were observed in the levels of FLV and ANT, which may be associated with protecting the photosynthetic machinery against excess solar radiation in conditions of reduced stomatal conductance [44]. The synthesis of FLV and ANT is associated with the protection of the photosynthetic apparatus from damage caused by reactive oxygen species (ROS) [45]; therefore, the production of these accessory pigments is associated with the mitigation of the harmful effects of drought in association with high solar radiation. Despite the increase in the synthesis of these accessory pigments, the nitrogen balance index (NBI), which relates the relative content of chlorophyll to the content of flavonoids, was less affected by the water conditions. However, for NBI, the most expressive reductions were observed in 2021, which may be associated with the lower intensity of water stress imposed in the 2021 crop season. In this crop season, for the same reduction in the concentration of chlorophylls estimated by the SPAD index, when compared to the 2020 CS, the concentration of FLV was higher, which is explained by the fact that when the chlorophyll content is divided by the FLV content, the relation becomes reduced, as shown in Figure 1.

The inexpressive reductions observed in the maximum quantum efficiency of photosystem II (Fv/Fm) show that even under severe water limitation, that is, at times when the soil reached values below the permanent wilting point, the photochemical machinery of popcorn plants was minimally compromised. Some authors report a certain tolerance of PSII to water limitation in the plant [46,47,48]. There were notable differences between the crop seasons for the Fv/Fm estimator; in 2020, the estimated values remained close to 0.75, even with a higher intensity of soil water limitation (25.4%), when compared to the 2021 crop season, in which the soil water limitation was 18%. In the 2021 crop season, even with lower water limitation, the Fv/Fm ratio values were higher, due to some impairment in the absorption of N since the limitation by N can reduce the maximum quantum efficiency of photosystem II.

The traits of the angle of brace (ABR) and crown (ACR) roots and density of brace (DBR) and crown (DCR) roots are associated with the formation of the root architecture in lateral and horizontal extension, cited by authors as key traits for the root ideotype; that is, phenotypes with steeper and deeper root systems [16,49,50,51]. The increases observed in ABR and ACR, when compared to the WW and WS conditions, are explained by the low water availability in the WS condition, a situation in which the plants develop steeper root systems, that is, the greater angles of growth concerning the surface of the soil are related to roots with greater ability to reach subsurface soil layers, to make such plants more tolerant of reduced rainfall [52,53]. The authors Gao and Lynch [54] and Kamphorst et al. [55], evaluating corn and popcorn genotypes, respectively, under drought conditions, reported increases of 17.05% and 23.71% for ACR, associating this increase in the magnitude of this trait to the genotypes more adapted to the drought condition [10,56].

The increase in DRB and DCR for the WS condition concerning WW in 2020 may be associated with a greater presence of secondary roots, which guarantees a greater surface for water absorption in a situation of soil water limitation [57]. Root phenotypes with smaller numbers and larger size of cortical cells reduce the metabolic cost of exploring deeper soil layers [54,55]. The crown roots are responsible for acquiring much of the water and mineral nutrients during the vegetative phase and are of great importance during the reproductive phase, when water stress is most critical [58]. The reductions in NBR (10.78%) and NCR (8.24%), especially in the 2021 CS, suggest a lower metabolic expenditure for the formation of the root system [12]. Such a reduction may be associated with the lower water limitation of the soil in 2021 (18%) compared to the greater water limitation that occurred in 2020 (25.4%). This fact denotes a root adaptation of the lines to the drought condition, in which a deeper system with smaller lateral extensions is observed for the most promising genotypes [58].

### 3.3. Identification of Superior Genotypes and Traits of Interest for Indirect Selection

Multivariate analysis (biplot GT) was implemented only with yield components, which, as a rule, are correlated with GY [59]. In addition to being highly correlated with GY, the selected traits show an effective response to the characterization of lines under drought conditions [10].

The genotypic superiority of L381, L292, L291, L688, and L321 in both CS and WC, especially for GY and PE, encourages the implementation of breeding programs aimed at obtaining superior popcorn genotypes for adaptation to environments with low availability of water. This group of lines can be used as parents, as they stood out for traits of high interest for genotypic resilience under water-limited conditions, specifically for their higher average estimates for PE, GY, 100 GW, and GNE, simultaneously. Breeding programs seek to work on selection by bringing together the most stable genotypes with the top yielded. Thus, L217 and L381 would be ideal to be used in a hybrid combination in the WS condition since, considering the scenario of water limitation already present, which has been increasing, these lines expressed a more stable response for the evaluated traits. This response was different from the L292 line, which, despite being considered the one with the highest yield, did not show significant stability. Regardless of CS, comparing the two water conditions, it is noted that the L291 line maintained good phenotypic stability under WS.

For expansion capacity (EP), using the L294, L688, L204, and L220 lines in the formation of hybrids is of fundamental importance, since the L294 and L688 lines have the highest phenotypic expressions for this trait. In addition, the L688 line was considered ideal for full irrigation conditions for traits related to grain yield. It is worth noting that there could be a harmful discrimination of L294 due to its expressive level of instability, both in the WS and WW conditions; however, this fact did not produce major adversities since grain expansion is the most explored trait of L294, which was not so affected by environmental changes when compared to traits related to grain yield.

In breeding programs, the discriminatory and representative capacity of a trait is extremely important, since it is possible to choose key traits to carry out a faster selection and obtain gains in other traits by indirect selection [60,61]. In this sense, for both WC and CS, the PH, GNE, and GY traits showed a potential for greater discrimination and representativeness, which favors the selection of lines with traits of high interest for adaptation to the soil water deficit condition. Grain yield is highly influenced by plant height (PH), relative chlorophyll content (SPAD index), and grain number per ear (GNE) and, based on these results, the high discrimination capacity of these traits indicates that the selection of genotypes based on these traits will result in increases in GY [62,63]. As contributions to the breeding aimed at the selection of popcorn lines, the GY, PE, 100 GW, and GNE traits stand out, which can be used in hybrid crosses for the formation of cultivars with high yield potential and adapted to the drought condition and to compose heterotic groups for the implementation of recurrent intra or interpopulation selection, as well as for the recycling of lineages aiming at the constitution of populations even more superior in terms of adaptation to drought.

## 4. Materials and Methods

### 4.1. Genotypes

Fifty popcorn lines from the Germplasm Bank of the State University of Northern Rio de Janeiro (Universidade Estadual do Norte Fluminense “Darcy Ribeiro”—UENF) were used in this research. The genealogy of these lineages derives from genotypes from Latin American countries with temperate and/or tropical climate adaptation (Santos et al., 2021), as described in Appendix A (Table A1).

### 4.2. Experimental Conditions and Crop Management

In the 2020 and 2021 crop seasons (CS), at Antonio Sarlo State Agricultural and Technical School (Colégio Estadual Agrícola Antônio Sarlo) (2°34′31″ S and 4°54′40″ W), in Campos dos Goytacazes, Rio de Janeiro, Brazil, the lines were cultivated under two contrasting water conditions, the well-irrigated condition (WW—“Well-Watered”) containing recommended irrigation for the crop, and the water stress condition (WS—“Water-Stressed”) in that the irrigation of the plants was suspended from 15 days before the male anthesis, lasting until the physiological maturation of the grains (harvest). The date of male anthesis was determined based on previous experiments [9,11,12].

The site has an automatic weather station that recorded the micrometeorological variables, which are shown in Figure 6. During the experimental period, the mean temperature was 21.77 °C and 21.26 °C, the mean relative humidity was 76.85% and 77.27%, and the photosynthetically active radiation was 1240.07 µmol m^−2^ s^−1^ and 1415.80 µmol m^−2^ s^−1^ for the 2020 and 2021 crop seasons, respectively (Figure 6).

The experimental design was randomized blocks with three replications. Each experimental unit consisted of a 4.40 m long row, spaced 0.20 m between plants and 0.80 m between rows, totaling 23 plants per plot. A drip irrigation system was implemented in the experimental area, with one dripper per plant, which promoted greater control of the amount of water applied. In the 2020 crop season, a total of 157.68 mm (WW) and 87.48 mm (WS) of water was applied via irrigation, that is, a reduction of 44.5% in the comparison between water conditions, and in the 2021 crop season, 132.93 mm (WW) and 78.92 mm (WS) of water was applied, a 50% reduction when comparing WW and WS (Figure 7). Incident precipitation was monitored, totaling 119.20 mm and 170.20 mm in the 2020 and 2021 crop seasons (Figure 7).

During the execution of the experiments, the soil water potential was monitored using Decagon MPS-6 sensors (DECAGON®, Pullman, WA, USA), which were installed between two plants located in the sowing row at a depth of 0.20 m. The plants received full irrigation in the WW condition, and the soil was close to the soil field capacity (−0.01 MPa). In the WS condition and the 2020 crop season, the soil reached the permanent wilting point (−1.5 MPa) at 63 days after sowing (DAS) (grain formation stage). In the 2021 crop season, the soil reached the highest water tension (−1.5 MPa) at 100 DAS (grain filling stage) (Figure 8).

Regardless of the crop season (CS) and water condition (WC), the sowing and topdressing fertilizations were conducted by providing 30 kg ha^−1^ N (urea), 60 kg ha^−1^ P_2_O_5_ (triple superphosphate), and 60 kg ha^−1^ K_2_O (potassium chloride). In the second topdressing fertilization, at 30 DAS, 100 kg ha^−1^ N (urea) were made available. In addition, weeds, pests, and diseases were controlled according to the needs of the crop.

### 4.3. Agronomic, Physiological, and Morphological Traits of the Root System Evaluated

The evaluated traits were divided into three groups: agronomic, physiological, and root system traits. The agronomic traits evaluated were the average plant height (PH), average ear height (EH), average stem diameter (SD), average ear length (EL), average ear diameter (ED), grain number per row (GNR), row number per ear (RNE), grain number per ear (GNE), 100-grain weight (100 GW), grain yield (GY), popping expansion (PE), and expanded popcorn volume per ha (EPV). PH and EH were measured after bolting, and PH was evaluated from the soil to the flag leaf, and for the variable EH, the measurement was made up to the insertion of the main ear. The average stem diameter (SD) was measured just below the main ear with a digital caliper (Digital Caliper Stainless Steel 150 mm Mtx). RNE and GNR were estimated by counting. PH, EH, SD, RNE, and GNR were estimated in a sample of ten plants, and the ears of these plants were randomly harvested. The 100 GW variable was quantified through two samples of one hundred grains from the plot after the ears were threshed. Grain yield (GY) was obtained after threshing the ears of each plot, which were corrected for 13% moisture (kg ha^−1^). The popping expansion (PE) was estimated by the mass of 30 g of grains irradiated in the microwave (1000 W) in a kraft type paper bag for 2 min, and the popcorn volume was quantified in a 2000 mL beaker; the ratio of the popped volume divided by 30 g determined the PE, which is expressed in mL g^−1^. EPV was obtained by multiplying GY and PE (m^3^ ha^−1^).

The physiological characteristics evaluated were the intensity of the green color of the leaves (estimate of the relative content of chlorophylls, SPAD index), anthocyanin index (ANT), flavonoid index (FLV), nitrogen balance index (NBI), maximum quantum efficiency of photosystem II (Fv/Fm). The evaluations were conducted during the grain filling period, 40 days after male flowering, in ten plants per plot, in the middle third of the leaf immediately above the main ear. To assess ANT, FLV, and NBI, the DUALEX equipment (DUALEX Scientific, Orsay, France) was used. The SPAD-502-Plus (SPAD 502, Minolta Company, Osaka, Japan) was used to estimate the leaf greenness (SPAD index). Chlorophyll fluorescence emission (Fv/Fm) was measured with a Pocket PEA handheld digital unmodulated fluorimeter (Hansatech Instruments Ltd, King’s Lynn, United Kingdom). This last trait was evaluated after dark adaptation of the area to be sampled for 15 min.

Root traits were measured according to the methodology proposed by [49], with modifications [24]. Regardless of WC, the root system of two plants per plot was removed in soil cylinders 25 cm in diameter and 40 cm in depth. The root traits measured were the number of brace roots (NBR), the number of crown roots (NCR), both estimated by counting; the density of brace roots (DBR), the density of crown roots (DCR), obtained using the scale proposed by [49]. Additionally, the angle of brace root (ABR), and the angle of crown root (ACR) were measured with the aid of a degree protractor and expressed in degrees (°) concerning the soil surface [49].

### 4.4. Statistical Analysis

The joint analysis of the years was based on the following statistical model: Y_ijkp = µ + 〖(B/WC)/CS〗_jkp + G_i + 〖WC〗_k + 〖CS〗_p + 〖GWC〗_(ik) + 〖GCS〗_ip + 〖WCCS〗_kp + 〖GWCCS〗_ikp + ε_ijkp; where: Y_ijkp = v observed value for the variable under study referring to the j-th repetition of the combination of the i-th level of the Genotype factor with the k-th level of the Water condition factor, with the p-th level of the Crop Season factor; µ = overall mean; 〖(B/WS)/CS〗_jkp = block effect; G_i = effect of the i-th level of the Genotype factor on the observed value Y_ijkp; 〖WC〗_k = effect of the k-th level of the Water Condition factor on the observed value Y_ijkp; 〖CS〗_p = effect of the p-th level of the crop season factor on the observed value Y_ijkp; 〖GWC〗_(ik) = effect of the interaction of the i-th level of the factor Genotype with the k-th level of the factor Water condition; 〖GCS〗_ip = effect of the interaction of the i-th level of the Genotype factor with the p-th level of the crop season factor; 〖WCCS〗_kp = effect of the interaction of the k-th level of the Water Condition factor with the p-th level of the crop season factor; 〖GWCCS〗_ikp = effect of the interaction of the i-th level of the Genotype factor with the k-th level of the Water condition factor with the p-th level of the crop season factor; and e_ijkp = error associated with the observation Y_ijkp, assuming NID (0, σ^2^).

The individual analysis was based on the statistical model: Y_ijk = µ + B_j + G_i + ε_ijk, where: Y_ijk = observed value for the variable under study referring to the j-th repetition of the combination of the i-th level of the Genotype factor; µ = overall mean; B_j = effect of the j-th repetition; G_i = effect of the i-th level of the Genotype factor on the observed value; ε_ij = experimental error associated with the observation Y_ij, assuming NID (0, σ^2^).

The multivariate analyzes “which won where/what”, “means vs. stability”, “discriminativeness vs. representativeness”, and “ranking genotypes” were performed considering the genotype by trait (GT) biplot model, using the standardized values of the variables strictly associated with the production components, that is, 100 GW, PE, GY, RNE, GNR, GNE, ED, and EL. To generate the GT biplot graph in the R software, the GGEbiplotGUI package was used.

## 5. Conclusions

The high phenotypic plasticity of the lines, given the significant interactions, meant that the experiments were analyzed individually, which represents an additional challenge for the popcorn breeder, as the breeding programs must be individualized for each water condition. The SPAD index, 100 GW, GY, PE, and GNE traits had the most significant impact on genotype selection. Regardless of WC and CS, the ideal lines were L294 and L688 for PE; L691 and L480 for GY; and L291 and L292 for both traits. SPAD index, 100 GW, and GNE can help in the indirect selection, independent of WC and CS. As contributions to popcorn breeding, the lines L294, L691, L291, and L292 are recommended to obtain hybrids or form heterotic groups, aiming to implement recurrent intra or interpopulation selection for the generation of superior genotypes with adaptation to the soil water limitation.

## Figures and Tables

**Figure 1 plants-11-02275-f001:**
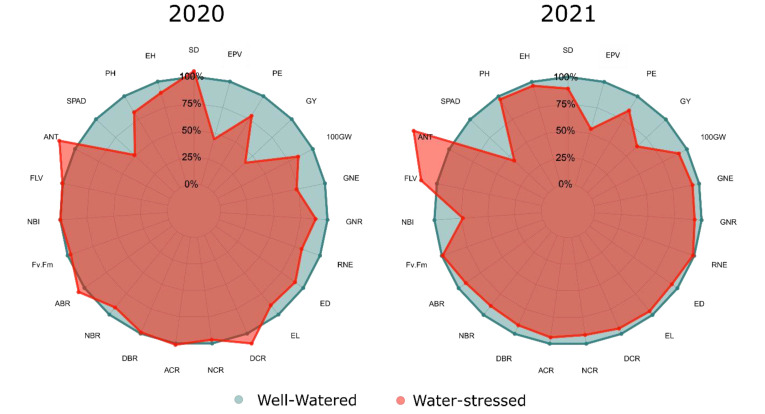
Percentage reduction between the averages of the traits evaluated in the different water conditions—Well-watered (blue) and Water-stressed (red)—in the 2020 and 2021 crop seasons (CS). EH—Ear height; PH—Plant height; SD—Stem diameter; EL—Ear length; ED—Ear diameter; RNE—Row number per ear; GNR—Grain number per row; GNE—Grain number per ear; 100 GW—100-grain weight; GY—Grain yield; PE—Popping expansion; EPV—Expanded popcorn volume; SPAD—Relative chlorophyll content; ANT—Relative anthocyanin content; FLA—Relative flavonoid content; NBI—Nitrogen balance index; Fv/Fm—Chlorophyll fluorescence; ABR—Angles of brace roots; ACR—Angles of crown roots; NBR—Number of brace roots; NCR—Number of crown roots; DBR—Density of brace roots; DCR—Density of crown roots.

**Figure 2 plants-11-02275-f002:**
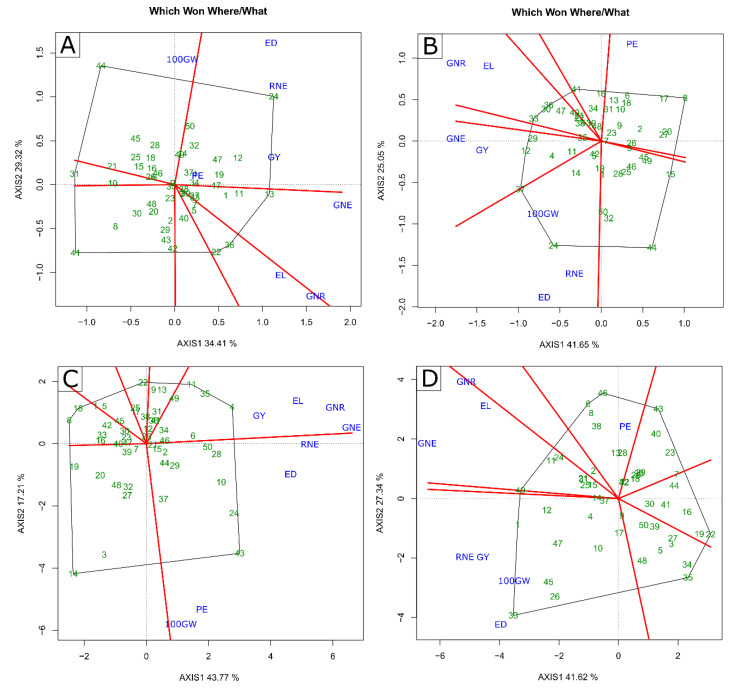
Biplot “which-won-where/what” graph for (**A**–**C**) water stressed (WS) and (**B**–**D**) well-watered (WW) conditions in 2020 (**A**,**B**) and 2021 (**C**,**D**) crop seasons. GY—grain yield; PE—Popping expansion; GNE—grain number per ear; 100 GW—100-grain weight; RNE—Row number per ear; GNR—Grain number per row; ED—Ear diameter; EL—Ear length; (-) signal indicating negative values.

**Figure 3 plants-11-02275-f003:**
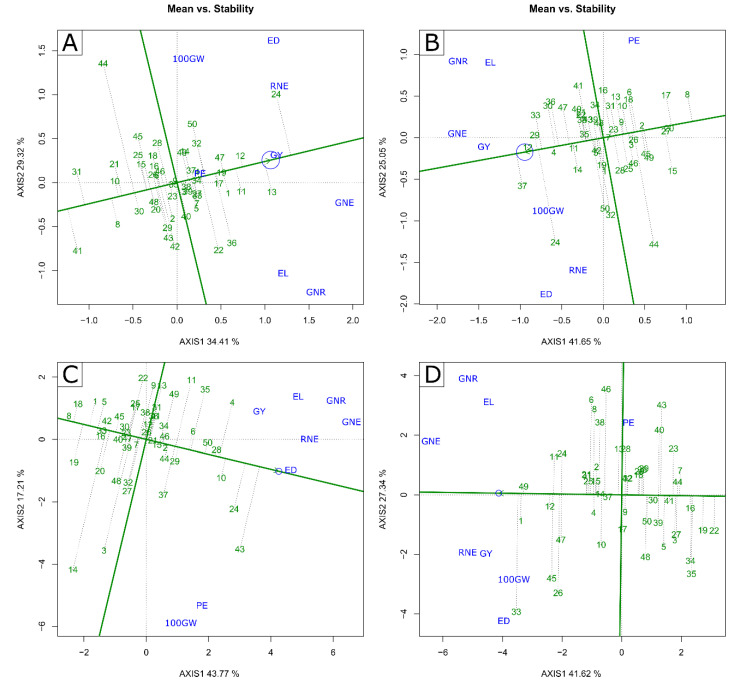
Biplot “Mean vs. Stability” graph for (**A***–***C**) water stressed (WS) and (**B**–**D**) well-watered (WW) conditions in crop seasons 2020 (**A**,**B**) and 2021 (**C**,**D**). GY—grain yield; PE—Popping expansion; GNE—grain number per ear; 100 GW—100-grain weight; RNE—Row number per ear; GNR—Grain number per row; ED—Ear diameter; EL—Ear length; (-) signal indicating negative values.

**Figure 4 plants-11-02275-f004:**
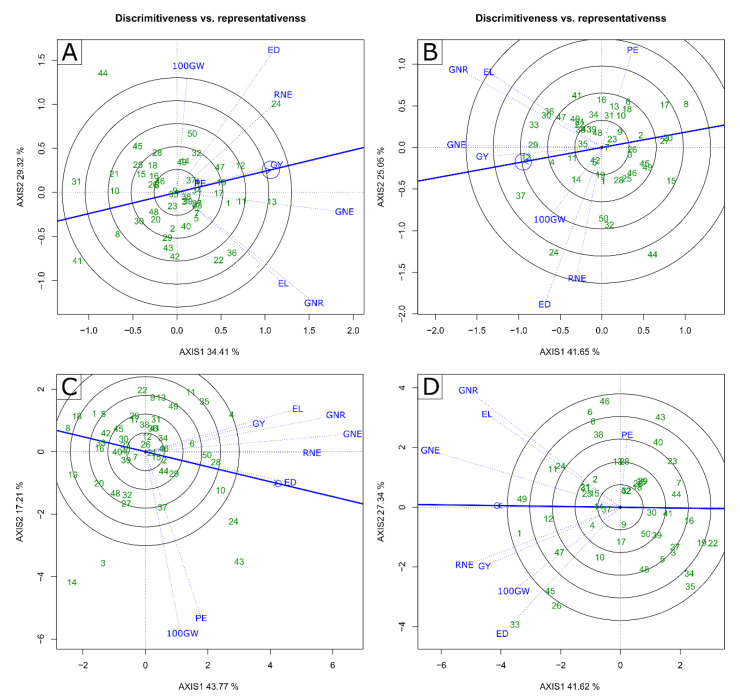
Biplot “Discrimitiveness vs. representativeness” graph (**A***–***C**) water stressed and (**B***–***D**) well-watered conditions in crop seasons 2020 (**A**,**B**) and 2021 (**C**,**D**). GY—grain yield; PE—Popping expansion; GNE—grain number per ear; 100 GW—100-grain weight; RNE—Row number per ear; GNR—Grain number per row; ED—Ear diameter; EL—Ear length; (-) signal indicating negative values.

**Figure 5 plants-11-02275-f005:**
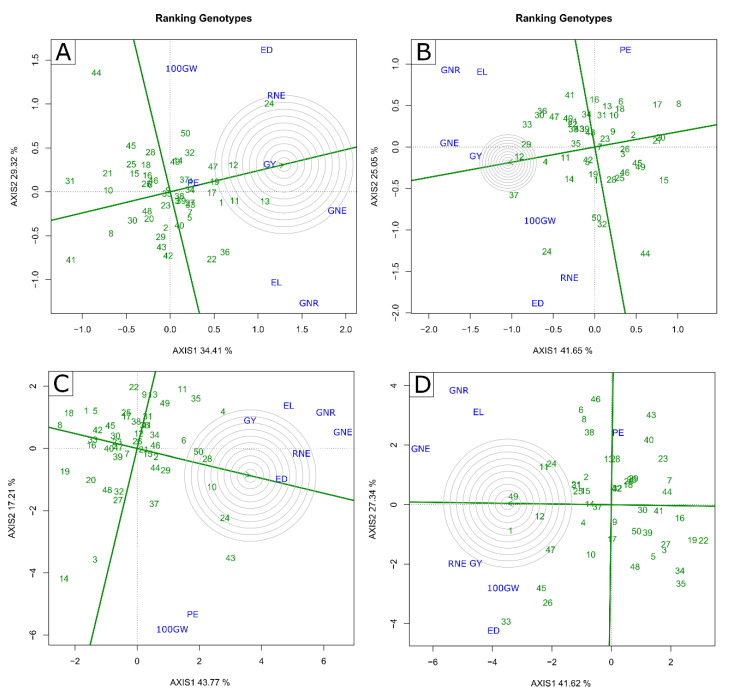
Biplot “Ranking Genotypes” graph for (**A***–***C**) water stressed (WS) and (**B***–***D**) well-watered (WW) conditions, in crop seasons 2020 (**A**,**B**) and 2021 (**C**,**D**). GY—grain yield; PE—Popping expansion; GNE—grain number per ear; 100 GW—100-grain weight; RNE—Row number per ear; GNR—Grain number per row; ED—Ear diameter; EL—Ear length; (-) signal indicating negative values.

**Figure 6 plants-11-02275-f006:**
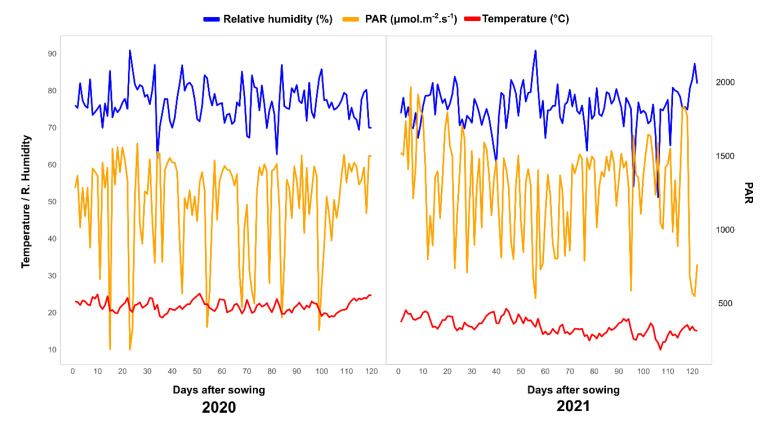
Environmental conditions (relative humidity, photosynthetically active radiation, and average temperature) observed in experiments with 50 popcorn lines cultivated in the 2020 and 2021 crop seasons.

**Figure 7 plants-11-02275-f007:**
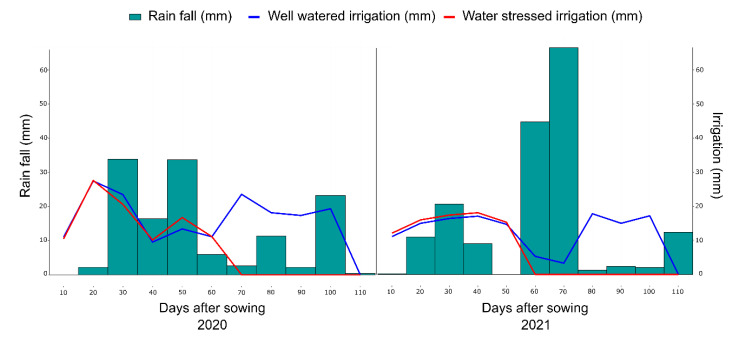
Amount of water applied (mm), in the 2020 and 2021 crop seasons, concerning the days after sowing of popcorn lines. WW (well-watered) and WS (water stress condition). The red line refers to the WS condition, and the blue line refers to the WW condition.

**Figure 8 plants-11-02275-f008:**
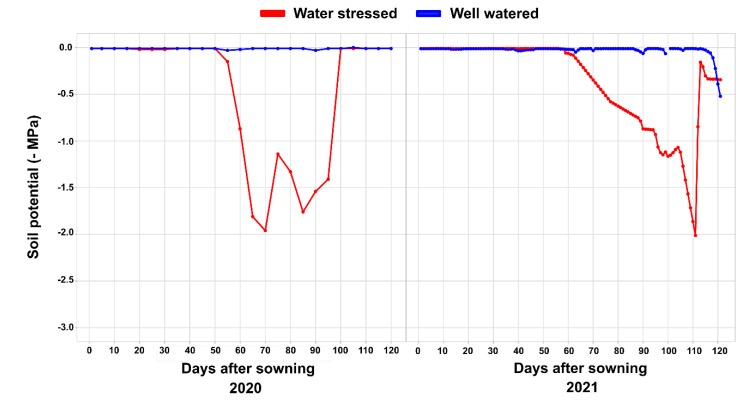
Soil water potential during the cultivation of popcorn lines under contrasting water conditions (WS and WW) in the 2020 and 2021 crop seasons.

**Table 1 plants-11-02275-t001:** Summary of joint and individual analysis of variance, estimates of means, and standard deviations associated with agronomic, physiological, and morphological characteristics of the root system of popcorn lines under different water conditions (WS and WW) and crop seasons (2020 and 2021).

Traits		Joint Analysis							WS				WW			
		CS	WC	G	CS*WC	CS*G	WC*G	CS*WC*G	2020		2021		2020		2021	
Agronomic	100-grain weight (g)	***	***	***	***	***	***	***	9.22 ± 1.39	**	9.15 ± 1.25	**	10.91 ± 1.60	**	10.00 ± 1.56	**
	Popping expansion (g·mL^−1^)	***	***	***	***	***	***	***	17.44 ± 3.90	**	18.76 ± 3.18	**	22.20 ± 3.87	**	22.16 ± 3.47	**
	Grain yield (kg·ha^−1^)	***	***	***	***	***	***	***	461.02 ± 200.62	**	812.08 ± 303.25	**	1519.64 ± 504.04	**	1285.05 ± 419.59	**
	Expanded popcorn volume (m^3^·ha^−1^)	***	***	***	***	***	***	***	8.07 ± 3.91	**	15.34 ± 6.47	**	33.46 ± 11.63	**	28.25 ± 9.62	**
	Row number per ear (unit.)	***	***	***	***	***	***	***	18.71 ± 2.86	**	22.54 ± 4.23	**	22.90 ± 3.21	**	22.79 ± 3.01	**
	Grain number per row (unit.)	**	***	***	***	***	***	***	11.64 ± 1.34	**	11.79 ± 1.30	**	13.11 ± 1.07	**	12.61 ± 1.22	**
	Grain number per ear (unit.)	**	***	***	***	***	***	***	220.07 ± 40.78	**	270.77 ± 70.05	**	299.63 ± 45.02	**	288.98 ± 54.43	**
	Ear diameter (mm)	***	***	***	**	***	***	***	25.18 ± 2.07	**	26.07 ± 2.13	**	27.79 ± 1.79	**	27.85 ± 2.37	**
	Ear length (cm)	***	***	***	***	***	***	***	9.72 ± 1.51	**	12.08 ± 1.39	**	10.96 ± 1.34	**	12.63 ± 1.33	**
	Ear height (cm)	***	***	***	***	***	***	***	81.08 ± 12.24	**	90.31 ± 11.52	**	90.90 ± 13.55	**	93.84 ± 10.08	**
	Plant height (cm)	***	***	***	***	***	***	***	128.94 ± 14.80	**	155.00 ± 16.68	**	156.38 ± 14.05	**	160.07 ± 15.47	**
	Stem diameter (mm)	***	***	***	***	***	***	***	12.91 ± 0.96	**	12.80 ± 1.43	**	12.28 ± 0.89	**	14.37 ± 1.33	**
Physiologic	SPAD index	***	***	***	***	***	***	***	15.14 ± 6.35	**	14.17 ± 4.06	**	29.75 ± 5.60	**	32.45 ± 5.54	**
	Anthocyanin index	***	***	***	***	***	***	***	0.27 ± 0.03	**	0.16 ± 0.03	**	0.23 ± 0.03	**	0.07 ± 0.03	**
	Epidermal flavonoids content	***	***	***	***	***	***	***	1.23 ± 0.12	**	1.10 ± 0.15	**	1.22 ± 0.11	**	0.95 ± 0.14	**
	Nitrogen balance index	***	***	***	***	***	***	***	20.14 ± 3.79	**	31.37 ± 6.06	**	20.12 ± 4.00	**	42.75 ± 9.13	**
	Maximum quantum efficiency of PSII (F_v_/F_m_)	***	***	***	*	***	***	***	0.73 ± 0.03	**	0.67 ± 0.05	**	0.76 ± 0.03	**	0.68 ± 0.04	**
Roots	Angles of brace roots (°)	***	^ns^	***	***	***	***	***	35.80 ± 5.92	**	42.52 ± 7.16	**	33.53 ± 4.96	**	46.37 ± 7.53	**
	Angles of crown roots (°)	***	**	***	***	***	***	***	41.91 ± 6.64	**	44.61 ± 8.29	**	41.36 ± 7.15	**	47.41 ± 6.59	**
	Number of brace roots	^ns^	***	***	^ns^	***	***	***	10.84 ± 2.30	**	10.76 ± 1.76	**	11.83 ± 2.20	**	12.06 ± 2.24	**
	Number of crown roots	***	***	***	*	***	***	***	16.24 ± 3.62	**	20.73 ± 3.56	**	16.82 ± 5.01	**	22.59 ± 4.57	**
	Density of brace roots	^ns^	***	***	^ns^	***	***	***	5.48 ± 0.89	**	5.31 ± 1.08	**	5.54 ± 0.90	**	5.81 ± 1.24	**
	Density of crown roots	***	^ns^	***	***	***	***	***	5.39 ± 0.73	**	4.72 ± 0.96	**	4.90 ± 0.90	**	4.99 ± 1.31	**

WS—Water stressed; WW—Well watered; CS—Crop season; WC—Water condition; G—Genotype; ***,** and * indicate a significant difference at the level of 0.01%, 1%, and 5% by the F test, respectively; ns: absence of significant difference by the F test at 5% probability.

## Data Availability

The data presented in this study are available in the article.

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
