# Peer review of "All Are in a Drought, but Some Stand Out: Multivariate Analysis in the Selection of Agronomic Efficient Popcorn Genotypes"

_plants, 2022, doi:10.3390/plants11172275_

Round 1
Reviewer 1 Report
This manuscript's authors conducted a two-year field experiment to identify the most drought-sensitive traits. It is an interesting study, and the field experiment is well designed. However, the description in the manuscript is somewhat confusing, with too many acronyms. And the description of the findings lacked details.
Line 22, "For CS", which CS?
Line 26, why all the components were increasing in 2021? Even under water stress conditions?
Line 22 - 27, the comparison is between WS and WW? If so, please be specific.
Line 29 -30, what are L196, L688, L691, L480, L291, and L292?
Line 101, what are ANT and FLV? Why do they increase under the WS condition?
Section 2, what are the general weather conditions in CS 2020 and 2021? Are they similar? Could the differences in weather conditions influence the results?
Line 176, "Regardless of WC..." but in the figure, it was separated by WC?
Line 178, what is L652 (44) ?
Figure 8, the data size of 2021 seems much larger than 2020. Why is that?
Reviewer 2 Report
The manuscript entitled “All are in a drought, but some stand out: Multivariate analysis in the selection of agronomic efficient popcorn genotypes” presents the possibility of using multivariate analysis to select promising popcorn lines and to identify yield components that could be used in the indirect selection of genotypes, under water-limited soil conditions.
The manuscript contains interesting research, is original and up-to-date as the subject, and is generally well structured and written. Based on the analysis of the manuscript, please find below some observations and suggestions, which could be used for the improvement of the work.
Title: I understand the desire of the authors to find an impactful title, but I think that a 'conventional' title would be preferable (simpler, clearer, and more relevant for readers). For example, it could be the last part of the current title: “Multivariate analysis in the selection of agronomic efficient popcorn genotypes” or an adaptation thereof.
Abstract: Review and try to simplify the Abstract.
Ensure a more concise and clear presentation of the abstract and comply with the journal's requirements (less than 200 words). In its current form, the Abstract is too extensive and especially with too many abbreviations and values. That's why it seems hard to follow. There is a risk of boring the reader who wants to quickly form an opinion about your research. In addition, because of the overly complicated abstract, the reader may no longer be interested in the full text of the manuscript. Try to remove abbreviations from the abstract or include them only after providing their meaning and following the instructions on the journal page (see "Abstract"): https://www.mdpi.com/journal/plants/instructions
Some sentences in Abstract (but also in the full text of the ms) are excessively long and the meaning of the results is not clear or is not explained (i.e., what does it mean “significant reductions”).
Please see:
Lines L 22-26: “For CS, significant reductions (>15%) between WC were observed in the values of 100-grain weight (100GW-15.50%), popping expansion (PE-21.45%), grain yield (GY-69.67%), expanded popcorn volume per ha (EPV - 75.89%), row number per ear (RNE-18.30%), plant height (PH-17.55%), and relative chlorophyll content (SPAD-49.11%) in CS 2020; and in PE (15.35%), GY (36.81%), EPV (45.70%), SPAD (56.34%), and nitrogen balance index (NBI - 26.62%) in CS 2021.”.
Are the percentage values of the reductions (between WS and WW?) for the analyzed traits in the Abstract relevant? It is not enough to say at the beginning that they are significant (>15%). And to explain which characteristics you are referring to, without values, because the reduction is over 15% for all of them.
Anyway, both in the Abstract and in the manuscript, you must clearly explain how you worked and why you consider the threshold of more than 15 significant for reductions (between WS and WW?), for example, what does it mean “significant reductions? (see above).
I suggest removing the years from the Abstract (reword the text so the calendar years don't appear; anyway, the years appear in the main text of the manuscript.). The validity of the research must be maintained over time, without seeming that because of the years it has become obsolete. Therefore, in L 18, instead of "2020 and 2021", you can write "in two consecutive years".
I suggest that you also give up the symbol from L 19, see '(ψ_soil≥ -1.5 MPa)'.
The Introduction and the following chapters are generally well written and the language is correct. Some too-long sentences could be simplified and formulated more clearly, to be easier for the reader to understand. Revise some terms, and notions or make small technical editing corrections (e.g., L 37 - use "CO2", instead of "CO2").
Check the order of the figures, so that both their citation and inclusion in the text are in chronological order (Figures 1, 2, 3, etc.). At this moment, the first figure cited in the texts is 'Figure 4' (see L 100).
Also, check the correspondence of the explanations in the text with the content of the figures and the values assumed in the text (which should be easy to observe or approximate in the figures). In this sense, check, for example, the correspondence between the figure - values in the following text:
Line L 96-100:
"In the 2020 crop season (24.5% reduction in irrigation water availability), significant decreases (> 15%) were observed among water conditions (WC), given the comparison between the averages found in WS concerning WW, based on the values of the 100GW (15.50%), PE (21.45%), GY (59.98%), EPV (65.89%), RNE (18.30%), PH (17.55%), and SPAD (49.11%) (Figure 4)."
Are you actually referring to Figure 1? Please revise carefully.
Regarding the issue of reduction reported already in the Abstract, respectively the percentage reduction between the averages of the traits evaluated in the different water conditions (well watered and water-stressed) in two successive crop seasons (CS), 2020 and 2021. Please see:
L 22-23: “significant reductions (>15%)..."
L 130: "Reductions in smaller intensities (< 15%)..."
L 101: "the greatest reductions (> 15%) (Figure 1)..."
L 136: "The root traits had little significant reductions (< 15%)..."
etc.
As previously mentioned, the threshold you refer to of 15 percent must be explained. You can do it in the 'Material and method' chapter, or if you have a bibliographical reference you can present the evaluation method in the 'Introduction' section, etc. But without adequate explanations, "significant reductions (>15%)" do not make sense for the readers, or at least for the readers who are not strictly in the scientific field.
In addition, avoid using the phrase "little significant reductions" (L 136), which is inappropriate scientifically, or at least from a statistical point of view.
Round 2
Reviewer 2 Report
The authors improved the manuscript according to the suggestions. I think that the manuscript can be accepted and can enter the editing process.
